# Partnering with Taxidermists for Improved Chronic Wasting Disease Surveillance

**DOI:** 10.3390/ani9121113

**Published:** 2019-12-11

**Authors:** Ashley Ableman, Kevin Hynes, Krysten Schuler, Angela Martin

**Affiliations:** 1Wildlife Health Unit, Wildlife Resources Center, New York State Department of Environmental Conservation, 108 Game Farm Road, Delmar, NY 12054, USA; kevin.hynes@dec.ny.gov; 2Cornell Wildlife Health Laboratory, Animal Health Diagnostic Center, Cornell School of Veterinary Medicine, 240 Farrier Road, Ithaca, NY 14853, USA; ks833@cornell.edu; 3Bureau of Environmental Exposure Investigation, Center for Environmental Health, New York State Department of Health, ESP Corning Tower, Albany, NY 12237, USA; angela.martin@health.ny.gov

**Keywords:** chronic wasting disease, prion, white-tailed deer, cervid, New York, taxidermist, transmissible spongiform encephalopathy

## Abstract

**Simple Summary:**

Chronic wasting disease (CWD) is a contagious neurological disease affecting deer, moose, elk, and reindeer. CWD is predominantly found in North America and has a higher prevalence in older male deer. To increase the submission of samples from older male deer, the Taxidermy Partnership Program (TPP) was implemented in New York State (NYS). This program partners with taxidermists to obtain valuable samples that would otherwise be lost and helps raise awareness about CWD. Since its start, the TPP has been successful in increasing the number of older male deer submitted for CWD testing.

**Abstract:**

Chronic wasting disease (CWD) is a neurodegenerative disease of cervids caused by a misfolded protein called a prion. This disease affects captive and free-ranging deer, moose, elk, and reindeer, and has been detected in 26 states. Cervids infected with CWD may be asymptomatic for months or years. In most areas, older male deer have higher prevalence rates. Prior to 2013, CWD surveillance in New York State focused on testing samples of convenience, by collecting deer heads from meat processors. However, this sampling was biased because many of the heads from older male deer were taken to taxidermists to be mounted. In 2013, the Taxidermy Partnership Program (TPP) was created to train taxidermists to collect CWD samples, and to increase the proportion of older male deer submitted for CWD testing. Added benefits include improved communication with taxidermists and increased awareness about CWD. Trained taxidermists were able to successfully collect and submit tissue samples with few errors. Participating taxidermists were paid for viable samples. Currently, there is a stable number of taxidermists that participate each year. This program has proven to be a valuable resource for obtaining high-value CWD samples for the wildlife agency, requiring a minimal amount of funding and time.

## 1. Introduction

Disease surveillance is a critical function in any wildlife health program. By knowing if a disease is present in an area, wildlife managers can either declare themselves free of disease (if not detected at a certain prevalence) or implement management actions to contain or eradicate the disease [1]. Surveillance samples are typically collected from the animals most likely to have the disease in order to create a more efficient sampling system. Adequate disease surveillance is key in the early detection of wildlife diseases. Without early detection, the disease may not be able to be contained. If left unchecked, human health and safety, as well as wildlife conservation efforts, may be at risk [2].

In recent years, chronic wasting disease (CWD) has emerged as a disease of international concern and represents a threat to cervid populations. CWD is a fatal prion disease of mule deer, white-tailed deer, elk, moose, and reindeer, predominantly found in North America, where it is currently spreading. Since its discovery in Fort Collins, Colorado in 1967 [3], CWD has been found in 26 US states and three Canadian provinces, as well as in South Korea, Norway, Finland, and Sweden.

As with most states in the eastern U.S., New York introduced a statewide CWD surveillance plan in response to the detection of CWD in Wisconsin in 2002. For wild deer, the New York State Department of Environmental Conservation (DEC) primarily collected hunter-harvested samples via discarded deer heads provided by meat processors [4]. In April 2005, the first positive case of CWD in New York was detected in five white-tailed deer from two captive breeding facilities in Oneida County. A subsequent containment area was implemented spanning Madison and Oneida Counties (Wildlife Management Unit 6P). After immediate intensive culling and testing of deer in the containment zone, the DEC detected CWD in two free-ranging white-tailed deer; a two-year-old doe and a near-yearling fawn. Mandatory testing and regulation of the containment zone ended in 2009. Since 2005, no additional cases of CWD were detected in either wild or captive deer in New York State (NYS). However, decreased sampling—from a high of 8000 samples in 2005, to less than 2000 samples in 2010—indicated that the DEC was devoting less effort to CWD sample collection.

Efforts by the DEC were renewed in 2012 to focus on prevention and early detection for effective management. The DEC continued statewide CWD surveillance annually, however, more emphasis was placed on testing any clinical deer reported by the public. Additionally, a pilot project was started to determine if taxidermists could act as “citizen scientists”, to successfully collect and submit samples for CWD.

Older deer (aged ≥2.5 years), especially older male deer, have a higher prevalence of CWD [5,6]. Thus, older males have a higher value in a weighted surveillance system that uses demographics to increase the probability of detection [7,8]. Prior to 2013, most samples tested during the hunting season came from meat processors, which included few older male deer. In 2013, the DEC’s Wildlife Health Unit (WHU) in collaboration with Cornell University’s Wildlife Health Laboratory created the Taxidermy Partnership Program (TPP). The objectives of the TPP included: (1) successfully train taxidermists to collect appropriate tissues (retropharyngeal lymph nodes); (2) increase the proportion of adult male deer in NYS CWD surveillance; (3) devise a payment structure that encourages taxidermist participation.

## 2. Materials and Methods

Our research for this manuscript did not cause harm to animals so I have not submitted an ethics approval.

Prior to implementation, DEC Wildlife staff gathered contact information for taxidermists statewide as there was no current master list or licensing requirements in NYS. To gauge CWD knowledge and interest, DEC contacted the United Taxidermists of New York and attended their annual meeting for two years. In 2013, the first year of the TPP, letters outlining the program were sent to NYS taxidermists soliciting participants in the sampling effort. After the first year, regional DEC biologists contacted their local taxidermists that expressed interest in cooperating to verify participation prior to the start of the hunting season in September. DEC biologists were given latitude to only select taxidermists they trusted to participate in this program.

Once participation was verified, sampling kits were assembled and sent directly to the taxidermist from the WHU. Each kit included: a data card, tissue vial, forceps, sterile disposable scalpel, and latex gloves. All deer tags and tissue vials are pre-labeled with corresponding barcode stickers. Originally, barcode stickers were included in the kits, but they were often improperly used by the taxidermists. To resolve this issue, the supplies are now sent pre-labeled. In addition to sampling kits, the taxidermists also receive written instructions, payment paperwork, and a NYS Wildlife Health Cooperator patch. To ensure proper training, a video created by DEC biologists at the WHU demonstrating sampling and packaging of the retropharyngeal lymph nodes (RPLN) was sent to all new taxidermists via DVD. The video is also available online: https://www.youtube.com/watch?v=Owpv30ulOvk.

When a new taxidermist joins the TPP, they are assigned a collector ID to match each sample with the appropriate taxidermist. The baseline payment for any processor or taxidermist with a collector ID is $5 (USD) for a deer of any age. An additional $5 is provided for each male deer ≥2.5-years-old. Appropriately collected RPLN samples qualify for an additional $5. To receive the full $15 payment for a sample, a taxidermist must submit the RPLN from a male deer ≥2.5-years-old, a completed data card, and a jaw bone to the DEC. DEC staff periodically check in with taxidermists to arrange sample pick up and supply more kits as needed. Jaw bones from each sample are aged via tooth wear and replacement [9] by WHU staff, and the RPLNs are sent to Cornell’s Animal Health Diagnostic Center for CWD ELISA testing.

## 3. Results

Our field survey of processors and taxidermists in operation yielded information on business status and practices that was not available elsewhere. There were 422 taxidermy businesses identified, but only 139 were confirmed to be open; 248 had unknown status and required additional contact. Of all the meat processing businesses, 223 of the 368 were confirmed as being in operation.

Despite the large numbers of taxidermists and processors in operation, only 17% of taxidermists and 25% of meat processors participate by providing deer heads or collecting RPLN. However, using the payment structure, we have been able to maintain a consistent number of participating processors and taxidermists (Figure 1).

Payments to participating taxidermists have not been excessive and likely are more cost-effective than other methods of agency sampling like clinical suspects (e.g., sending a biologist out to collect a reported sick or dead deer) or obtaining heads from a meat processor (e.g., biologist visits meat processor, centralized sample collection, and shipment to the diagnostic laboratory). The average payment to taxidermists was $270 with a pay range from $15 to $885. Meat processors were paid an average of $168.

For the past 6 years, we recorded whether samples were collected by a DEC biologist or a taxidermist. The correct tissues (lymph nodes) were collected by both groups 99% of the time. Incorrect tissues were collected infrequently and usually were confused with salivary glands. Taxidermists had better sample collection rates than biologists in five of the six years (Figure 2), but there was no statistical difference between the two groups (*p* = 0.6).

Using taxidermists to collect samples has increased the proportion of adult males being tested in relation to the total number of deer tested for CWD (Figure 3). Prior to the program, in 2010, the percentage of male deer ≥2.5-years-old tested for CWD was 15.6%, but increased to 34.3% in 2018.

## 4. Discussion

As CWD continues to spread across the country, it is important to investigate new avenues of efficient and effective surveillance. We found that taxidermists were an untapped source of valuable CWD samples and could be successfully trained with minimal agency effort. Taxidermists were just as capable in collecting the appropriate tissues and following submission instructions for laboratory testing as staff biologists. Due to the age and sex demographic of deer received by taxidermists and their success in tissue extracting, they have proved to be a reliable source of high-value CWD samples. These samples would otherwise be lost to surveillance without the TPP in place. With an average of 23 taxidermists, NYS has been able to increase the sampling proportion of older age class males by 18.7% from 2010 to 2018 (Figure 3).

While the cost to have taxidermists collect samples might seem relatively expensive if considered on a per-sample basis, it is likely on par with or cheaper than agency collection if time and costs of sample extraction, transportation, and storage are considered. It costs the agency ~$90 per sample when staff time and processor payment is taken into consideration. Although taxidermists are typically paid more, using taxidermists to extract samples cuts staff time down considerably. The consistency of taxidermist participation indicates the current payment system is a good incentive for tissue collection.

The TPP has created a citizen science program that allows relevant business owners to actively participate in defending New York against CWD. Having programs like the TPP also raises awareness in the taxidermist and hunting community about CWD. The benefits of the TPP greatly outweigh the cost of paying taxidermists, as valuable tissues are collected appropriately with the training material provided. In-person interaction between a DEC biologist and taxidermists develops better relationships between the agency and its stakeholders. We also used this opportunity to provide educational materials that taxidermists could then distribute to other hunters. Additionally, there are a number of regulatory changes and voluntary recommendations associated with CWD. Taxidermists are sent notices of regulatory changes by the agency as they occur. Voluntary recommendations are sent periodically to provide taxidermists with best management practices to avoid contaminating their business and landscape with CWD. Established citizen science programs, like the TPP, can create better relationships between stakeholders and state agencies, particularly when businesses are concerned about regulations that may affect their operations, such as bans on the interstate import of whole deer carcasses.

## 5. Conclusions

Since its implementation, the Taxidermy Partnership Program has proved to be a cost-effective method for obtaining high-value samples for CWD testing. The taxidermists participating in the program have demonstrated that they can successfully identify, collect, and submit appropriate tissue samples. The established payment process in place has kept consistent interest and participation in the program. Due to the success of the program, New York has increased the proportion of high-value older male deer submitted for chronic wasting disease surveillance, without considerable added effort or funding.

## Figures and Tables

**Figure 1 animals-09-01113-f001:**
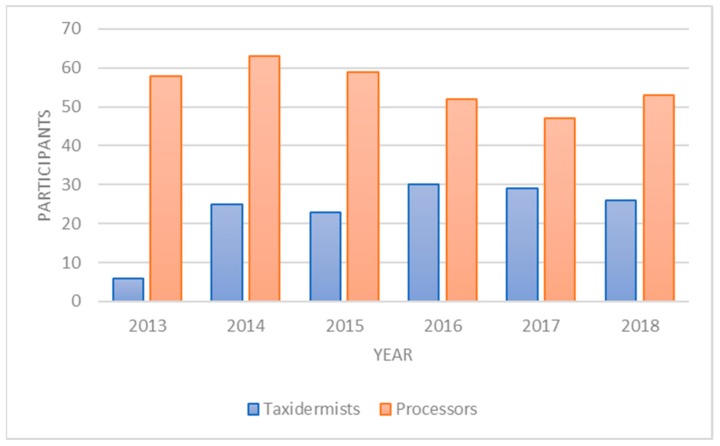
Taxidermist and processors providing deer heads or collecting retropharyngeal lymph node (RPLN) samples for Chronic Wasting Disease (CWD) testing in New York State since 2013.

**Figure 2 animals-09-01113-f002:**
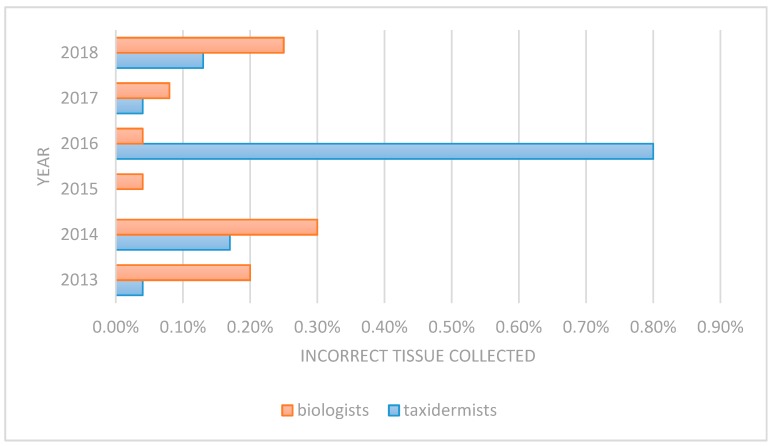
The percentage of incorrect tissues collected by both taxidermists and Department of Environmental Conservation (DEC) biologists has remained below 1% since the start of the Taxidermy Partnership Program. The most common reason for sample rejection is that the incorrect tissue was collected, such as the salivary gland.

**Figure 3 animals-09-01113-f003:**
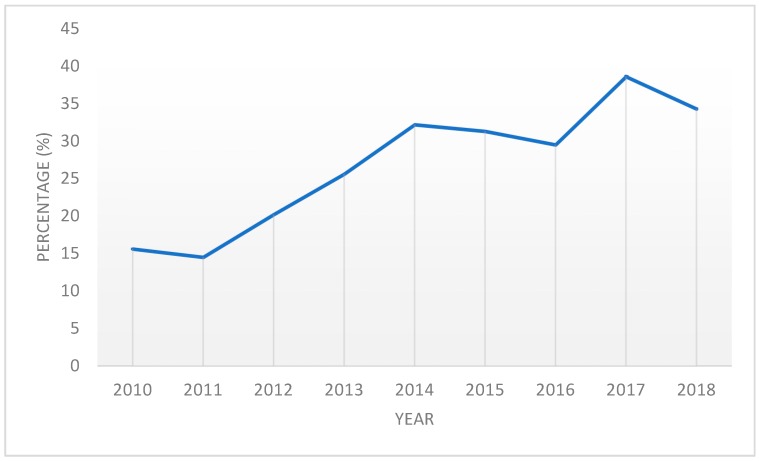
Since the start of the Taxidermy Partnership Program (TPP) in 2013, the percentage of adult male deer submitted for surveillance has increased.

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
