# Peer review of "Partnering with Taxidermists for Improved Chronic Wasting Disease Surveillance"

_animals, 2019, doi:10.3390/ani9121113_

Round 1
Reviewer 1 Report
Manuscript ID: animals-644173
Title: Partnering with Taxidermists for Improved Chronic Wasting Disease Surveillance
Authors: Ashley Ableman, Kevin Hynes, Krysten Schuler, Angela Martin
Reviewer Summary:
This manuscript describes the efforts taken by the New York State Department of Environmental Conservation to increase the submissions of adult male deer tissue samples for chronic wasting disease (CWD) testing. The authors focus on the partnership they have developed with state taxidermists to collect these samples from hunter-harvested deer in the state. This partnership has led to a notable increase in adult male deer sample submissions at what seems to be a reasonable cost to the state.
Reviewer comments:
This manuscript is fairly well-written and provides some insight into just one mechanism that state wildlife agencies can use to increase the number of submissions from deer which may be more likely to carry CWD. There are several small but important areas which the authors may wish to focus on to improve the manuscript, as follows:
1) Materials and Methods lines 86-87: It may be helpful for the readers to restate the years of activity, e.g. “In the first year of the TPP, 201X [2013 I believe?], letters outlining the program…”
2) Materials and Methods lines 101-106: This paragraph is somewhat confusing because of the sentencing structure. E.g. “An additional $5 each is provided for male deer…” might read better as “An additional $5 is provided for each male deer…”. The next sentence states that an additional $5 is provided for RPLN pulled by a taxidermist. It’s unclear what’s meant here – does it imply that if the RPLN were pulled by someone other than a taxidermist, the $5 would not be received, or does it imply that if something other than an RPLN were sampled, the $5 would not be received? Perhaps a better way to convey the meaning would be:
“Processors or taxidermists with a collector ID are reimbursed up to $15 for each submission. A baseline reimbursement of $5 is provided for all submissions that include a completed data card and a jawbone, regardless of age or sex of the animal. Appropriately collected RPLN samples qualify for an additional $5, as do samples provided from adult males over 2.5yrs old.”
3) Results lines 115-116: The authors state that a relatively small percentage of taxidermists and processors participate in the partnership. It may be helpful, in addition to Figure 1, to state the range of percentages from each group providing samples over the project years. The phrase “a relatively small percentage” begs the question in the reader’s mind “what is that percentage?”
4) Figure 2 is…confusing, and feels a bit like filler material. What may be more appropriate is to simply show the average payouts to taxidermists along with a range. The authors may also consider showing the payouts to processors, and including their contribution in the title of the manuscript.
5) Results lines 142-144: There is a misspelling “adult males being testing,” and some general confusion in what has increased. Is the percentage of submissions from male deer over 2.5yrs of age increasing in proportion to all the deer tested? Or is it increasing in proportion to the number of adult male deer over 2.5yrs of age that are harvested? I believe it is the proportion of all deer tested, but this should be clarified.
6) Discussion lines 157-158: In conjunction with the previous comment, the authors state in the Discussion “…NYS has been able to increase the number of older age class males by 18.7% from 2010 to 2018.” Once the authors have better defined what has increased, it may be more appropriate to say that “NYS has been able to increase the sampling proportion of older age class males by 18.7%...”
7) Discussion lines 159-161: While it’s acceptable to speculate that “the cost to have taxidermists collect samples…is likely on par or cheaper than agency collection…”, what would really helpful in boosting the utility of the manuscript is to try to break down those agency costs. As with comment #3 above, the statement begs the question in the reader’s mind “how much does it cost to have the agency collect the samples?”
8) Discussion lines 170-171: This sentence again feels like filler material. Are these regulatory changes and voluntary recommendations provided to the taxidermists? Are they provided in the educational materials to be distributed to hunters, referenced in the previous paragraph? It would be helpful to better characterize, or flesh out, this comment.
These comments are, for the most part, minor, though providing responses would benefit the message that the authors are trying to convey. I appreciate the opportunity to review the manuscript and look forward to the authors’ revisions.
Reviewer 2 Report
This ms provides information about gaining CWD samples from taxidermists. While there is merit in the approach, the ms has significant lapses in logic, unsubstantiated statements, and missing information that detract from the value of the paper. The data have merit, but the current presentation is lacking,
Author Response
Response to Reviewer 2 comments:
Point 1: This ms provides information about gaining CWD samples from taxidermists. While there is merit in the approach, the ms has significant lapses in logic, unsubstantiated statements, and missing information that detract from the value of the paper. The data have merit, but the current presentation is lacking,
Response 1: Your comments have been taken into consideration and the manuscript has been updated. The author's would like to thank you for your review.